# Optimization of Processing Steps for Superior Functional Properties of (Ba, Ca)(Zr, Ti)O_3_ Ceramics

**DOI:** 10.3390/ma15248809

**Published:** 2022-12-09

**Authors:** Cristina Elena Ciomaga, Lavinia P. Curecheriu, Vlad Alexandru Lukacs, Nadejda Horchidan, Florica Doroftei, Renaud Valois, Megane Lheureux, Marie Hélène Chambrier, Liliana Mitoseriu

**Affiliations:** 1Department of Exact and Natural Sciences, Institute of Interdisciplinary Research, Al. I. Cuza University of Iasi, Carol I, 700506 Iasi, Romania; 2Dielectrics, Ferroelectrics & Multiferroics Group, Faculty of Physics, Al. I. Cuza University of Iasi, Carol I, 700506 Iasi, Romania; 3“Petru Poni” Institute of Macromolecular Chemistry, Grigore Ghica Voda Alley 41 A, 700487 Iasi, Romania; 4UMR 8181—UCCS—Unité de Catalyse et Chimie du Solide, Univ. Artois, CNRS, Centrale Lille, Univ. Lille, F-62300 Lens, France

**Keywords:** BCTZ, synthesis and sintering routes, dielectric properties, electrocaloric effect, dc-tunability, piezoelectric response

## Abstract

Lead-free piezoelectric ceramics with nominal composition at morphotropic phase boundary Ba_0.85_Ca_0.15_Ti_0.9_Zr_0.1_O_3_ (BCTZ) prepared by different processing routes and sintered either by conventional solid-state reaction or by spark plasma sintering (SPS) techniques were comparatively investigated to observe the role of structural modifications and of microstructures on the dielectric, ferroelectric, piezoelectric and electrocaloric responses. The ceramics presented relative densities from 75% to 97% and showed variations in their phase composition as a result of variable mixing and different synthesis and sintering parameters providing local compositional heterogeneity. As result, all of the ceramics showed diffuse phase transition and ferroelectric switching responses, with parameters affected mostly by density (*P_r_* between 3.6 to 10.1 μC/cm^2^). High values for the electrocaloric response in the Curie range were found for the ceramics with predominantly orthorhombic character. Field-induced structural modifications were probed by tunability anomalies and by XRD experiments in remanence conditions. Piezoelectric effects with notably high figure of merit values were assigned to the better densification and poling efficiency of BCTZ ceramics.

## 1. Introduction

Piezoelectric materials are currently used in many applications, such as sensors and actuators, due to their ability to generate electric charges by the application of mechanical inputs or strain under the application of an electric field. For more than 50 years, the lead zirconate titanate (PZT) family remained the prototype of piezoelectric ceramics due to its large piezoelectric coefficient (*d*_33_) and temperature stability. Besides PZT, the other piezoelectrics usually employed in applications belong to the category of Pb-based relaxors. However, the toxicity and environmental impacts of lead limited its use and led to the development of a new class of electroceramics, i.e., lead-free ceramic oxides. Among the potential candidates as lead-free materials, BaTiO_3_—based solid solutions seem to have great potential as piezoelectrics, but also as high permittivity ferroelectrics for multilayer ceramic capacitors, tunable applications or electrocaloric elements. However, as piezoelectrics, they are characterized by relatively smaller values of *d*_33_ (~300 pC/N, i.e., about half of the value reported for the PZT family). The basic approach to achieve high piezoelectricity is to place the composition of such solid solutions in the proximity of a multi-phase range or “morphotropic phase boundary” (MPB), where the polarization direction can be easily rotated by external stress or an electric field, resulting in a high piezoelectric and dielectric response [1,2]. For a long time, the MPB of Pb-based systems was considered “superior” in terms of providing better piezoelectricity to those that resulted in Pb-free ferroelectrics, without a reasonable explication. A few years ago, W. Liu and X. Ren [3], and then D.S Keeble et al. [4], showed that the phase diagram of (1 − *x*)Ba(Zr_0.2_Ti_0.8_)O_3_ − *x*Ba_0.7_Ca_0.3_TiO_3_ (BZT-BCT or BCTZ) contains a *phase convergence region around x = *0.5**, where different amounts of all of the crystalline polymorphs (cubic-rhombohedral-orthorhombic-tetragonal) may coexist, similarly as for Pb-based systems with compositions in the range of MPB [5]. In this region, the energy barrier of polarization rotation and lattice distortion is low, thus allowing high piezo-, pyro-, and ferroelectric responses.

The functional properties (dielectric, tunability, piezo-, pyro-, and ferroelectric characteristics) of systems with compositions around their MPB are usually strongly dependent on the structural phase composition, which is determined by processing parameters [6] and, therefore, a similar feature was expected in BCTZ ceramics with phase superposition. Due to such effects, a large variety of functional properties and broad ranges for the material constants’ values have been reported for BCTZ ceramics with composition around MPB in different studies [7,8,9]. 

For a specific composition at MPB, the polymorph amounts and the functional properties can be tailored by using different routes and variable parameters for (i) *the powders synthesis*: solid state reaction [10], sol-gel method [11], oxalate co-precipitation [12], citrate method [13], hydrothermal [14] or microwave-assisted synthesis methods [15], reactive template grain growth [16]; and by (ii) modifying the *sintering techniques* or the control parameters in conventional [17], spark plasma [18], hot pressing sintering [19], etc. Besides the variable processing methods and parameters, modifications of phase compositions around MPB can be induced by electrical inputs [20,21,22]. The application of such external driving forces may change the free energy local minima and stabilize one or more polymorphs at the expense of the others, as shown by the Landau-based approaches and experimentally demonstrated in both Pb-based and Pb-free ferroelectric solid solutions [23,24].

In that regard, the present work aimed to investigate the influence of the structural modifications and microstructural parameters on the ferro-/piezo-/dielectric properties measured under different field sequences in a series of Ba_0.85_Ca_0.15_Ti_0.9_Zr_0.1_O_3_ ceramics produced by a few different processing routes (synthesis and sintering). The paper provides a comparative study of the dielectric characteristics under low and high electric fields, ferroelectric properties, dc-tunability, piezoelectric response, and electrocaloric effect for BCTZ ceramics obtained by different synthesis processes and sintered by conventional solid-state reaction method or spark plasma sintering (SPS) techniques. The effect of poling on the structural characteristics was observed, along with a corresponding discussion concerning its role in the tunability and piezoelectric properties. This research contributes to understanding the behavior of BCTZ ceramics under various stimuli and explores the possible processing routes providing enhanced functional responses.

## 2. Sample Preparation and Experimental Details

Five different types of Ba_0.85_Ca_0.15_Ti_0.9_Zr_0.1_O_3_ (BCTZ) ceramics were produced by the solid-state method, starting from the same precursors: BaCO_3_ (Sigma Aldrich), CaCO_3_, ZrO_2_, and TiO_2_ (Strem Chemicals). The reagents were weighed in stoichiometric proportions and then mixed by using different milling parameters and thermal treatment methods for calcination, as follows: (1) dry mixing of precursors for 30 min. by manually milling in agate mortar, followed by a calcination step at 1000 °C for 8 h (samples denoted as **BCTZ1**); (2) wet ball milling of reagents for 30 min at 350 rpm in a planetary mill (RETSCH S100), followed by a calcination step at 1000 °C for 4 h (denoted as **BCTZ2**); (3) wet-mixing of the precursors by using a vibration milling machine (RETSCH MM400) for 30 min at a frequency of 30 Hz, followed by a calcination step at 1000 °C for 4 h (denoted as **BCTZ3**); (4) wet-mixing of the precursors by using a vibration milling machine for 1 h at a frequency of 30 Hz, followed by a calcination step at 1000 °C for 4 h (denoted as **BCTZ4**); and (5) mixing of the precursors in a planetary mill (RETSCH S100) for 30 min at 350 rpm, followed by two calcination steps, at 1200 °C for 24 h and then at 1350 °C for 6 h, with intermediate manual milling steps (denoted as **BCTZ5**).

After calcination, the Ba_0.85_Ca_0.15_Ti_0.9_Zr_0.1_O_3_ powders (BCTZ1-BCTZ4) were de-aggregated by wet manual milling in an agate mortar, pressed into pellets at 1400 bar hydrostatical pressure using an Isostatic Press P/N 28640000, sintered in atmospheric air at 1450 °C for 2 h with a heating rate of 5 °C/min, and then slowly cooled (12 h) down to room temperature using a Nabertherm L5/13/P330 furnace. BCTZ5 powder was densified by spark plasma sintering (SPS-630SX machine) at 1300 °C and 18.8 kN pressure for 10 min. 

The structural analysis of the calcinated powders and sintered ceramics was realized with an X-ray diffraction (XRD) analyzer (Shimadzu LabX 6000 diffractometer with Ni-filtered CuKα radiation, *λ* = 1.5418 Å). The samples mounted in reflection mode were measured at ambient atmosphere with a scanning step of 2°/min and a counting time of 0.6 s over the 2θ = (20–80)° range. The surface morphology of the BCTZ samples was examined by scanning electron microscopy (SEM) with a Scanning Electron Microscope type Quanta 200 (FEI), operating at 20 kV with secondary electrons in low vacuum mode (LFD). The ceramic density was estimated by using Archimedes’ method, and data are shown in Table 1.

The electrical measurements were realized on parallel-plate capacitor configurations, by applying Pd—Ag electrodes on the polished surfaces of the sintered ceramic disks. The complex impedance in the frequency domain of 100–2 × 10^5^ Hz and at temperatures of 20–160 °C was determined by using an Agilent E4980A Precision LCR Meter. The hysteresis *P*(*E*) loops measured at different temperatures were recorded by using a ferroelectric tester (Radiant Technologies: Precision Premier II, High Voltage Amplifier 10 kV). The high voltage dc-tunability measurements were performed at room temperature on electrode ceramic disks immersed in transformer oil, at high voltages produced by a function generator coupled with a high-voltage amplifier (TREK 30/20A-H-CE, Trek Inc., Medina, NY, USA) and a custom-designed circuit, as described elsewhere [25]. Piezoelectric coefficient *d*_33_ was determined by using a Piezometer Piezotest PM300, after poling at variable *dc* electric fields of 5 and 10 kV/cm for 10 min at room temperature. After each piezoelectric test, the ceramics were refreshed by heating at 600° for 2 h.

## 3. Results and Discussions

### 3.1. Structural and Microstructural Analysis of Ceramics

The XRD patterns of the calcinated powders and sintered Ba_0.85_Ca_0.15_Ti_0.9_Zr_0.1_O_3_ ceramics are represented in Figure 1a–f, and they indicate the formation of a BCTZ perovskite structure. The XRD patterns of BCTZ calcinated powders also showed the presence of a few unidentified small peaks at 2θ~28.4°, 37.5°, and 54.4°, while the reflection at 2θ~28.4° corresponded to unreacted BaZrO_3_ (Figure 1a) [4,26,27,28], which disappeared after the sintering step. The detailed representation of these diffractograms around 2θ~45° (Figure 1b), where the (002)/(200) tetragonal perovskite peaks were expected (Figure 1c), and around 2θ~75° corresponding to the (103)/(310) reflections (Figure 1d), indicated the phase co-existence for the Ba_0.85_Ca_0.15_Ti_0.9_Zr_0.1_O_3_ series with nominal composition at the morphotropic phase boundary (MPB). After the thermal treatment for densification, i.e., by sintering at 1450 °C for 2 h, the XRD patterns revealed the formation of pure BCTZ perovskite without secondary phases for all investigated ceramics, under the detection limit of XRD analysis (Figure 1d). The exact determination of phase composition was impossible using state-of-the-art laboratory XRD analysis, in particular for BaTiO_3_-based systems at MPB, where more phases with small crystallographic distortion usually coexist. There are still debates even concerning the room temperature phase symmetry of the well-known and extensively investigated pure BaTiO_3_ [29,30,31,32], which usually shows a tetragonal structure at room temperature. However, some indications concerning the predominant polymorph could be observed by the examination of the peak intensity ratio at 2θ~45° (where (002)/(200) tetragonal reflections were present). The I_002_/I_200_ ratio determined from the experimental patterns of polycrystalline tetragonal piezoceramics is usually about ~0.58, close to the multiplicity ratio of 0.50 characterizing ceramics with randomly oriented domains [33,34,35]. This peak intensity ratio can be modified by field/stress-induced preferential domain orientation as a surface effect (surface texturing) [36,37,38], but if the ceramic surface is not modified by processing or by field action, it may be an indication of a volume effect. In such a case, the different I_002_/I_200_ ratios will show the presence of polymorphs other than tetragonal (*T*) (such as the orthorhombic (*O*) state), or possible phase superpositions, particularly if the peaks are overlapped and broadened [29]. All of the calcinated powders and sintered ceramics in this study showed I_002_/I_200_ far from 0.5 or even >1, indicating their multi-phase character at room temperature, which was expected for the Ba_0.85_Ca_0.15_Ti_0.9_Zr_0.1_O_3_ nominal composition [39]. The calcinated powders BCTZ2, BCTZ4, and BCTZ5 presented a I_002_/I_200_ ratio <1, so they were characterized by a predominantly tetragonal phase, while BCTZ1 had a I_002_/I_200_ ratio >1, thus being more orthorhombic, and BCTZ3 showed a broad feature at 2θ~45°, characterizing a multi-phase character. After sintering, only BCTZ2 remained predominantly tetragonal, with a subunit ratio of I_002_/I_200_ <1, while the BCTZ1 ceramic was described by a superposition of polymorphs with a highlighted phase belonging to the orthorhombic structure (Figure 1e), and the ceramics denoted as BCTZ3, BCTZ4, and BCTZ5 presented a broad splitting at 2θ~45° (Figure 1e) and at 2θ~75° (Figure 1f) around the diffraction reflection of (103)/(310) planes that showed triple diffractions, indicating mixed phase multiplicity, such as those reported in the literature for similar compositions [3,4,28,40]. They still have a ratio of I_002_/I_200_ >1, meaning that orthorhombic symmetry might be predominant among the other possible structures. The differences between BCTZ1 and BCTZ3–5 could be related to the result of the relaxation of internal stress existing in the ceramic volume in the BCTZ ceramics with lower density in comparison with the dense ones. With respect to this, for BaTiO_3_ ceramics, a tendency towards the orthorhombic state was clearly promoted in denser ceramics, compared to the tetragonal state found in more porous ceramic structures [41,42]. However, the ceramic BCTZ2 is predominantly tetragonal, although it is also porous with the same porosity level as BCTZ1, meaning that not only the porosity of the final ceramic product but also the properties of the starting powders determine the phase symmetry or phase coexistence. In fact, among the corresponding calcinated powders, one was obtained by dry manual milling followed by calcination at 1000 °C for 8 h (BCTZ1) and was predominantly orthorhombic, while one produced by wet milling in the planetary mill and calcined at 1000 °C for 4 h (BCTZ2) was predominantly tetragonal, and this explains the differences in the XRD patterns of the corresponding sintered ceramics. In conclusion, the presented structural data obtained by laboratory XRD for the investigated BCTZ set of ceramics with composition around MPB were not enough to clearly and quantitatively establish the phase composition, and only detailed studies based on high-energy synchrotron X-ray analyses completed with structural calculations would be necessary [4,9,26,43]. As is known from the literature, the phase diagram of the BCTZ system has remained controversial ever since its emergence, mainly because the small structural distortions of perovskites are very difficult to detect and consequently, reports have been published on the co-existence of various combinations of rhombohedral (*R*), *O* and *T*-phases for the same composition: Ba_0.85_Ca_0.15_Ti_0.9_Zr_0.1_O_3_ [3,4,44]. 

Scanning electron microscopy (SEM) micrographs performed on the fresh fractured surfaces illustrate the microstructural features of BCTZ ceramics sintered by conventional (Figure 2a–d) and Spark plasma sintering methods (Figure 2e). As one can observe in Figure 2a,b, the ceramic samples BCTZ1 and BCTZ2 obtained by manual and ball milling of the precursor’s powders are less dense than the other ones. Indeed, their relative density determined by the Archimedes method by considering a theoretical density of 5.687 g/cm^3^ [45] are similar for BCTZ1 and BCTZ2, of 75% (BCTZ1) and 76% (BCTZ2) and strongly increases for the other samples, reaching 96% (BCTZ3), 95% (BCTZ4) and 97% (BCTZ5). All the BCTZ ceramics exhibit large grains with an average grain size of around 10–20 µm, including the BCTZ5 ceramic sintered by SPS by using powders subjected to two calcination steps at 1200 °C for 24 h and 1350 °C for 6 h, with intermediate manual milling steps.

### 3.2. Comparative Analysis of Low-Field Dielectric Properties

The real part of permittivity and dielectric losses measured vs. temperature, at 10 kHz frequency (Figure 3a,b), indicated the presence of the ferroelectric–paraelectric phase transition by a well-defined broad peak around the temperature of ~100 °C. With respect to the room temperature dielectric constant ranging between 2000–3000, an increase in permittivity to a value of 6500 at *T_m_* = 97.5 °C (for BCTZ1) and to 12,120 (for BCTZ5), with a small shift in the Curie temperature at 102.5 °C (for BCTZ5), was observed (Figure 3a). In the case of ceramic samples BCTZ2–4, conventionally sintered at 1450 °C for 2 h, the maximum value of permittivity presented an increase of up to 11,500 and a slight shift in the Curie temperature from 97.5 °C down to 87.5 °C. The significantly low value of permittivity in BCTZ1 can be attributed to the effect of porosity. All of the noticed variations in permittivity values, the aspect of permittivity vs. temperature dependences, and the transition temperature values could be assigned to the differences in phase composition, density, and local compositional heterogeneity obtained with the different synthesis and calcination methods employed to produce the various types of ceramic samples BCTZ1–5. It is worth noting that even though BCTZ1 and BCTZ2 had similarly low densities, their dielectric constants were quite different, i.e., the values corresponding to BCTZ1 were the smallest among all of the ceramic samples, while the permittivity of the BCTZ2 ceramic (76% relative density) was quite similar to (and the maximum even higher than) that of the BCTZ3 ceramic (characterized by a relative density of 96%). Even the exact amount of polymorphs was impossible to determine; the inversion of (002)/(200) peak intensity indicated that these ceramics had different phase compositions that would have a strong impact on their dielectric properties. All of the ceramics presented low dielectric losses (tan δ < 3% at all investigated temperatures) with a maximum followed by a strong drop around the phase transition (Figure 2b). As a general trend, the ceramics sintered by the conventional method presented lower losses (<2%) than the ceramics densified by the SPS method. In the case of ultrafast sintering processes such as SPS, some ionic species may not be fully equilibrated in the system, the presence of oxygen vacancies is likely, and such charges may contribute to an increase in dielectric losses by hopping mechanisms [18]. This can explain the higher losses characteristic of the BCTZ5 ceramics with respect to the others observed in Figure 2b.

Considering that all of the ceramics presented a diffuse phase transition due to the phase superposition, possible local variations in composition (in particular, the Zr/Ti stoichiometry that resulted from the different types of precursors mixing), and porosity in the case of the BCTZ1 and BCTZ2 ceramics, a Curie–Weiss modified analysis [46] was performed for the dielectric permittivity data determined at the frequency of 10 kHz. The obtained empirical parameters are listed in Table 1. It can be observed that the lowest Curie–Weiss temperature *T*_0_ corresponded to the ceramic with a predominantly orthorhombic phase (BCTZ1). The *η* parameter that gives information about the diffuseness of the phase transition (*η* = 1 characterizes a normal Curie–Weiss behavior, while *η* = 2 describes a completely diffuse phase transition) was around 1.6 for all of the ceramics, which means that all of them had a broad phase transition due to the mentioned effect of phase superposition. The highest Curie constant describing the strongest ferroelectric interaction was associated with the BCTZ5 ceramic (which also presented the largest permittivity values overall in the temperature range) and the BCTZ3 ceramic, as a result of an excellent densification, while the smallest values were associated with the BCTZ1 and BCTZ2 ceramics, due to the dilution of the ferroelectric character by the presence of a large number of pores. 

### 3.3. High Field Properties (Ferroelectricity, Tunability, and Piezoelectric Effect)

Figure 4 shows the *P*(*E*) hysteresis loops measured at room temperature for a sine wave input field with a frequency of 1 Hz and 15 kV/cm amplitude. All of the loops were well saturated for this maximum applied field, presenting a remanent polarization (*P_r_*) between 3.6 and 10.1 μC/cm^2^, and a similar coercive field in the range of 2–3 kV/cm, with a slight reduction from BCTZ1 to BCTZ5 (Table 1). These values are in good agreement with those reported in the literature for similar BCTZ ceramics [47,48]. The reduced values for the remanent and saturation polarizations and the aspect of tilted loops observed for ceramic samples BCTZ1 and BCTZ2 were results of the higher porosity of these samples [49]. On the other hand, the slight differences observed in the ferroelectric characteristics of samples BCTZ3–5, which had almost the same density, can be attributed to the mentioned possible differences in their polymorphic composition. The largest remanent *P_rem_* and saturation *P_sat_* polarizations for BCTZ4 can be explained by considering the predominant *O* phase characterizing this ceramic [28]. 

To evaluate the role of the different processing methods in the electrocaloric effect (EC) in the set of ceramics with similar densities (BCTZ3-5), the *P*(*E*) hysteresis loops were recorded at different temperatures (Figure 5a–c). As expected, all of the loops presented a clear diminution of remanent polarization with increasing temperature, due to the gradual reduction, until its disappearance, of the contribution from ferroelectric domains. The EC was calculated by an indirect method, using the upper branches of the hysteresis loops, by integrating the Maxwell relation [50]:(1)$$∆T=-\frac{T}{\rho C_{E}}\int_{E_{1}}^{E_{2}}{\left(\frac{\partial P}{\partial T}\right)}_{E}dE$$
where *ρ* is the ceramic’s density, *C_E_* is the specific heat capacity at a fixed field, and *E*_1_ and *E*_2_ represent the limits for the applied electric fields. In the present case, these were considered to be *E*_1_ = 0, *E*_2_ = 12.5 kV/cm, *ρ* = 5.476 g/cm^3^ (BCTZ3), 5.402 g/cm^3^ (BCTZ4), and 5.527 g/cm^3^ (BCTZ5), while *C_E_* was considered to be the same for all ceramic samples: of 0.378 J/gK, according to the data reported in Ref. [48]. 

Figure 5d shows the calculated dependences of the electrocaloric temperature change ΔT vs. temperature for all of the dense ceramic samples. Sample BCTZ4 had the largest temperature variation ΔT = 0.56 K at 353 K (near its Curie range), while BCTZ5 had the smallest effect, but showed a splitting feature that confirmed in this way the fact that this type of ceramic is characterized by a larger local compositional inhomogeneity. This is probably due to the fact that SPS is a fast sintering process that does not allow complete ionic interdiffusion like in the other ceramics produced by traditional sintering. Moreover, it seems that the specific processing employed to prepare the perovskite powders (two calcination steps with intermediate manual milling) did not provide enough compositional homogeneity. It is worth noting that the present EC was larger than others reported in the literature, which were obtained under even higher fields [48,49], thus showing that by using the synthesis and processing parameters employed for BCTZ4, optimization of this property can be reached. Usually, the electrocaloric responsivity (ζ = ΔT/ΔE) is more suitable for the evaluation and comparison of various materials for EC applications (Table 2). For the present set of ceramics, ζ had values of 0.34 Kmm/kV (BCTZ3), 0.45 Kmm/kV (BCTZ4) and 0.24 Kmm/kV (BCTZ5), which represent values among the highest reported in the literature for BCTZ ceramics [47,48,51,52]. 

The high polarization response and the higher values for the EC effect observed for BCTZ4 and BCTZ3 could be related to the predominant role of the orthorhombic polymorph in these ceramics. A higher polarization is usually assigned to the orthorhombic state due to the 12 possible polarization orientations available in this state, compared to only six in the tetragonal phase. 

The high-field dependence of permittivity (dc-tunability) was determined at room temperature under complete increasing/decreasing dc-cycles, and the results are shown in Figure 6a–e. All of the compositions presented strong variation in permittivity with the applied field, with a small hysteresis behavior and without any tendency toward saturation for the available maximum applied field. A very interesting feature was noticed for all of the samples, which is more evident for ceramics sintered by conventional sintering (BCTZ1–4) around a critical field placed in the range ±*E*~(3.75–4) kV/cm (indicated by the arrows in the graphs): BCTZ1 showed a hysteretic *ε*(*E*) dependence at low fields, and around this critical field, became almost non-hysteretic. BCTZ2 was non-hysteretic below this field and showed hysteresis above this field, while BCTZ3 and BCTZ4 presented a hysteretic *ε*(*E*) behavior, but at the critical field, the loops were pinched. This behavior was stable and well-reproducible, and it was found after many increasing/decreasing field cycles separated by a few (at least three) refreshing thermal treatments at 600 °C for 2 h. Such a peculiar feature was previously observed in the case of BaTiO_3_ ceramics with different grain sizes showing polymorph phase superposition around room temperature [32]. In systems with metastable polymorphic states for compositions placed around MPB, such as in the present case, field-induced structural transformations are likely, and they are the origin of such anomalies detected in the dc-tunability data. The tunability of these ceramics (*n* = *ε*(0)/*ε*(*E*)) varied from 2 for BCTZ4 to 1.45 for BCTZ1 at the field of *E* = 12 kV/cm. Even the tunability values showed two interesting features: (i) the largest value was found for the ceramic BCTZ4, where the predominant *O* phase supposed to be present in this ceramic played an important role in improving all of the high field properties [24]; (ii) BCTZ2 and BCTZ5 had the same values of tunability for all of the investigated fields; even the density and dielectric permittivity of BCTZ2 were much lower. This behavior can also be explained by considering the phase composition: BCTZ2 was more tetragonal at room temperature than BCTZ5, whose XRD pattern (Figure 1d,e) seemed to indicate a mixture of polymorphic phases. These observations show that another tool for tailoring the tunability properties in bulk ceramics around the MPB is to modify their phase composition by changing the synthesis and processing parameters [24]. 

### 3.4. Structural Modifications Induced by Poling

It was shown that BCTZ1 and BCTZ2 (with similarly low densities of 74% and 76%) had quite different low and mostly high-field properties. These features were assigned to their different diffractograms qualitatively indicating different phase compositions, i.e., BCTZ1 was more orthorhombic than BCTZ2, which appeared more tetragonal (Figure 1d,e). Since the high field properties are strongly affected by the phase composition, a structural study on the poled ceramics at different applied fields was performed, in order to observe, in remanence, structural modifications induced by the field application at a qualitative level. In order to eliminate the effect of density on the phase composition [41], the results for the dense ceramics only are presented in Figure 7a–f. The poling process was realized by using the following procedure: different dc electric fields of 5, 10, 15, and 20 kV/cm were applied for 10 min, at room temperature. The poled samples were aged for 24 h, and then XRD analysis was performed in remanence, at room temperature. After every applied electric field and measurement of the structural evolution, the samples were refreshed at 600 °C for 2 h. BCTZ1 in its virgin state had a predominant *O* phase, and after the application of fields with amplitude of 5 and 10 kV/cm, the intensities of peaks (002), (200), (103), and (310) were almost the same, while for higher fields, the *O* phase again became predominant (Figure 7a–c). 

The non-poled BCTZ2 sample presented a majoritarian *T* phase structure (Figure 7d–f). For an electric field lower than 5 kV/cm, the XRD diffractogram revealed a reduction of *T* phase, with a decrease and change in the intensity of peaks (002), (200), (103), and (310) compared to the unpoled BCTZ2 sample. By applied a poling field of 10 kV/cm, the BCTZ2 sample exhibited features of the coexistence of tetragonal and orthorhombic phases. For poling fields larger than 10 kV/cm, the field seemed to stabilize the orthorhombic phase. Therefore, it was noticed that a poling electric field of ~10 kV/cm induced pronounced structural changes with respect to the structure of the unpoled BCTZ2 ceramic. In contrast to the BCTZ1 ceramic, which showed a 75% relative density similar to that of BCTZ2, the second sample prepared by using wet ball milling for 30 min at 350 rpm in a planetary mill indicated that it was more responsive to the poling electric fields. It is worth mentioning that these results are only qualitative, being obtained with a state-of-the-art diffractometer. Therefore, they did not reach enough resolution for performing structural refinements, and further high-energy diffraction experiments are necessary for a detailed quantitative analysis. Nevertheless, these qualitative data allow us to conclude that for the BCTZ system at MPB, the applied electric field induces structural changes (e.g., modifying the *O*-*T* relative content or even inducing new polymorphs), and such an effect clearly contributes to modifications of the high field properties. 

### 3.5. Piezoelectric Properties

Considering the observed structural modifications and their influence on the high field properties of BCTZ ceramics with MBP composition, a piezoelectric study was also performed. The piezoelectric response for the investigated BCTZ ceramics was recorded on samples poled at room temperature for 10 min, in a manner similar to that previously described. Figure 8a shows comparative data for piezoelectric responses in all of the BCTZ ceramics, which were subjected to poling electric fields of 5 and 10 kV/cm, respectively. It could be seen that the piezoelectric coefficient (*d*_33_) was higher when increasing the applied field and when the ceramic density was higher. A maximized *d*_33_ value of ~283 pC/N was found for dense (95–97% r.d.) BCTZ4 and BCTZ5 ceramics, compared with only ~150 pC/N in the porous BCTZ1 (75% r.d.) sample, and such values of *d*_33_ were also reported for the BCTZ composition (e.g., *d*_33_ ≅ 285~424 pC/N presented in [53,54] or *d*_33_ ≅ 355pC/N in ref. [55]). This increase in the piezoelectric response was related to the higher fraction of the piezoelectric phase, which was correlated also with the increase in the remanent polarization (Figure 4), and was a combined contribution from the easy polarization rotation and domain wall motion afforded by the structural heterogeneity of BCTZ with composition at MPB [9,56]. 

In order to describe the piezoelectric performances in the BCTZ ceramics, the capability for mechanical energy harvesting applications was also estimated by computing the piezoelectric figure of merit (*FOM*) by using Equation (2) [57]:(2)$$FOM_{33}=d_{33}^{2}/\varepsilon_{33}^{T}$$
where $d_{33}$ represents the piezoelectric strain coefficient, and $\varepsilon_{33}^{T}$ is the relative permittivity. 

As can be seen from Figure 8b, the values of *FOM*_33_ were increased from ~5 (pC/N)^2^ for BCTZ1 to ~27.5 (pC/N)^2^) for BCTZ4. A similar observation was reported in [56,58], and these values are higher than those reported in other studies concerning similar BCTZ ceramic compositions [53,54]. 

## 4. Conclusions

In conclusion, the synthesis methods are determinants of the phase composition, dielectric properties, ferroelectric *P*(*E*), electrocaloric effect, tunability, and piezoelectric responses of Ba_0.85_Ca_0.15_Ti_0.9_Zr_0.1_O_3_ ceramics. The crystallographic structure and microstructural characteristics are influenced by the processing steps used for the fabrication of such BCTZ ceramics. The temperature dependence of the dielectric constant indicated an increase in permittivity and a small shift in Curie temperature related to the coexistence of polymorph phase, density, and local compositional heterogeneity. The investigation of the electrocaloric effect revealed that sample BCTZ4 presented the largest temperature variation ΔT = 0.56 K at 353 K (near its transition) and electrocaloric responsivity of 0.45 Kmm/kV, which was a larger value than reported in the literature, showing that by optimizing the processing route, an enhanced electrocaloric effect can be achieved. The high polarization response and the higher values for the EC effect observed for BCTZ4 and BCTZ3 were related to the predominant role of the orthorhombic phase in these ceramics. The dc-tunability data demonstrated that the existence of the *O* phase in the BCTZ4 ceramic played an important role in improving the high field properties. Therefore, another approach to tailoring tunability properties in ceramics with composition at MPB is to modify the synthesis and processing parameters. The detailed investigation concerning the structural modification induced by poling in the dense BCTZ ceramics indicated field-induced polymorph transformations, also confirmed by anomalies in the *ε*(*E*) dependences. The improved ferroelectric, dc-tunability, and piezoelectric properties of the investigated BCTZ samples were related to the different coexisting phases near room temperature, this composition being sensitive to the applied external fields, stress, and processing routes.

## Figures and Tables

**Figure 1 materials-15-08809-f001:**
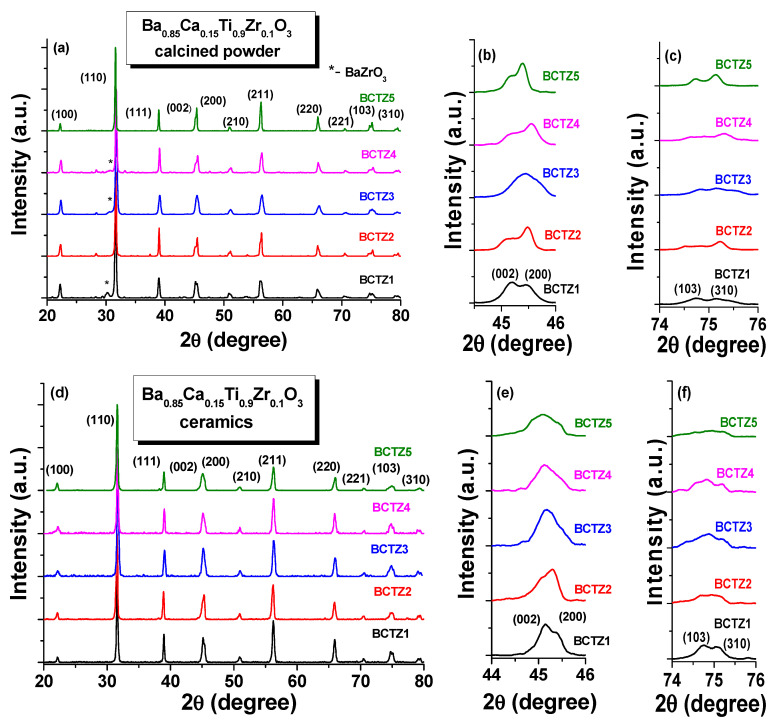
XRD patterns for BCTZ (**a**) calcinated powders and (**d**) ceramics (the enlarged views of the peaks around (**b**,**e**) 2θ~45° and (**c**,**f**) 2θ~75).

**Figure 2 materials-15-08809-f002:**
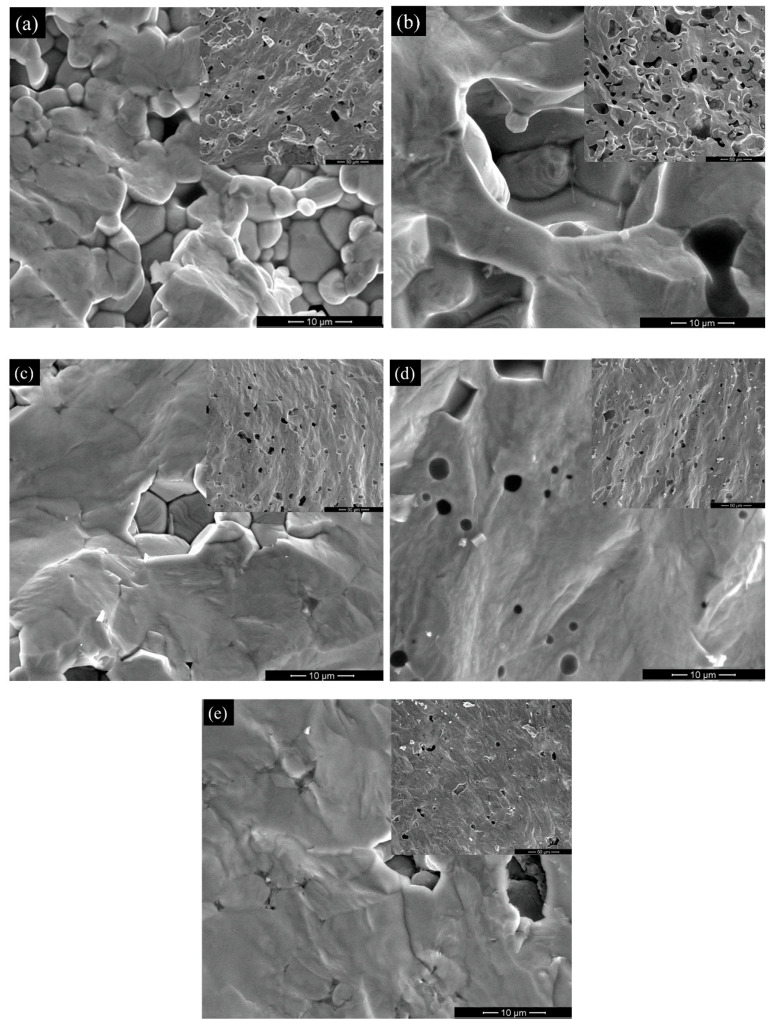
SEM micrographs of (**a**) BCTZ1 (75% r.d.), (**b**) BCTZ2 (76% r.d.), (**c**) BCTZ3 (96% r.d.), (**d**) BCTZ4 (95% r.d.) obtained by conventional sintering, and (**e**) BCTZ5 (97% r.d.) by Spark Plasma Sintering.

**Figure 3 materials-15-08809-f003:**
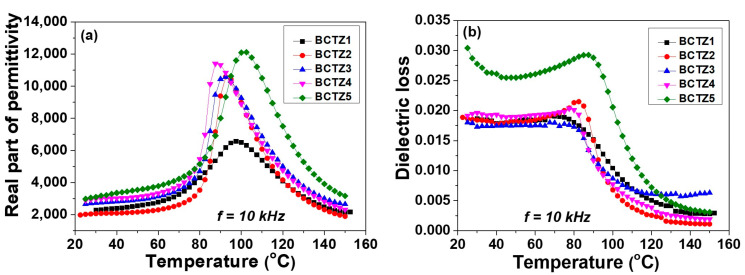
Real part of the dielectric constant (**a**) and tangent loss vs. temperature (**b**) for all of the BCTZ ceramic samples, measured at 10 kHz frequency.

**Figure 4 materials-15-08809-f004:**
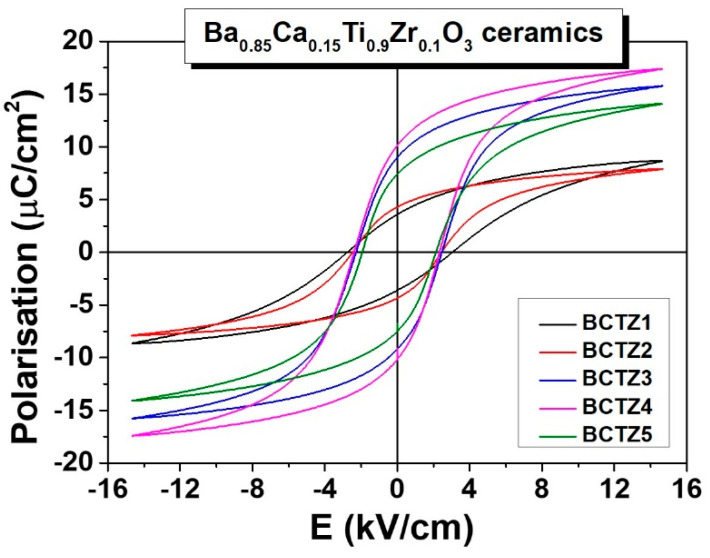
*P*(*E*) hysteresis loops of BCTZ ceramics determined under an input cycling field (sine wave) with the amplitude of 15 kV/cm and the frequency of 1 Hz.

**Figure 5 materials-15-08809-f005:**
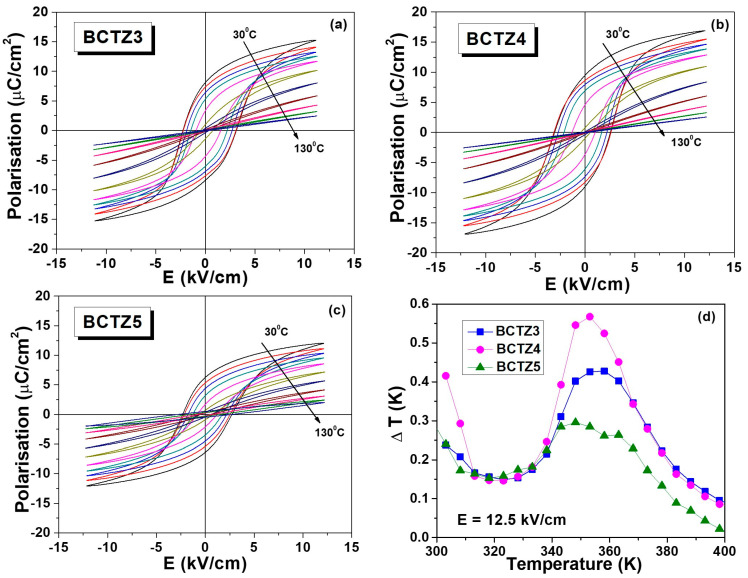
(**a**–**c**) *P*(*E*) loops at several selected temperatures for the BCTZ dense ceramics: (**a**) BCTZ3, (**b**) BCTZ4, (**c**) BCTZ5 and (**d**) the temperature dependence of the electrocaloric temperature change (ΔT) under field variation ΔE = 12.5 kV/cm.

**Figure 6 materials-15-08809-f006:**
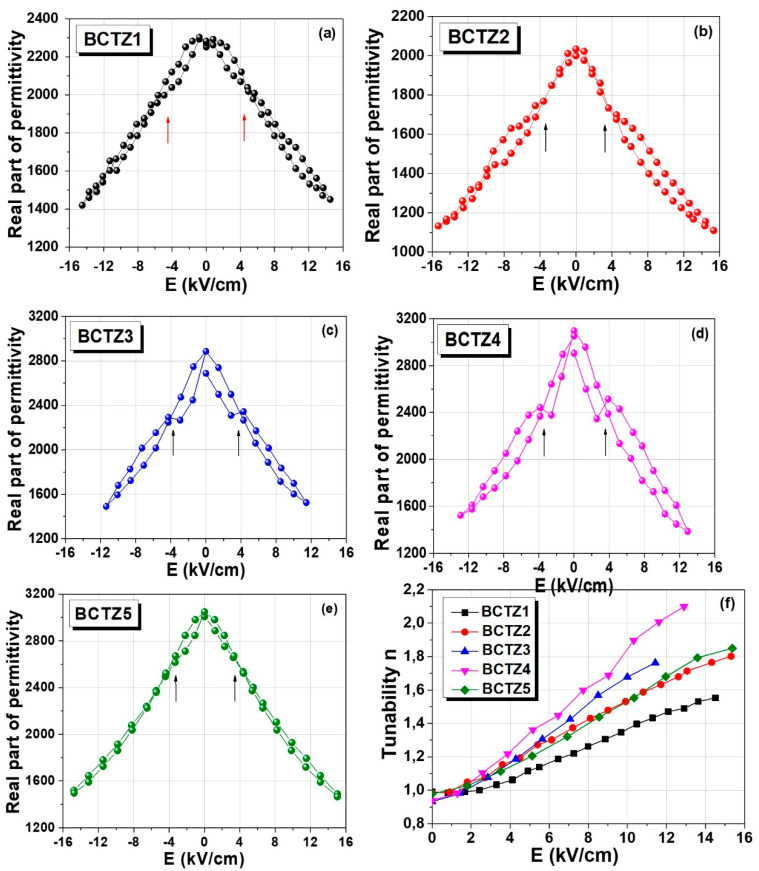
Permittivity vs. dc-electric field under an increased/decreased field cycle (**a**–**e**) and comparative tunability dependence on the dc-electric field (**f**) for all of the BCTZ ceramics.

**Figure 7 materials-15-08809-f007:**
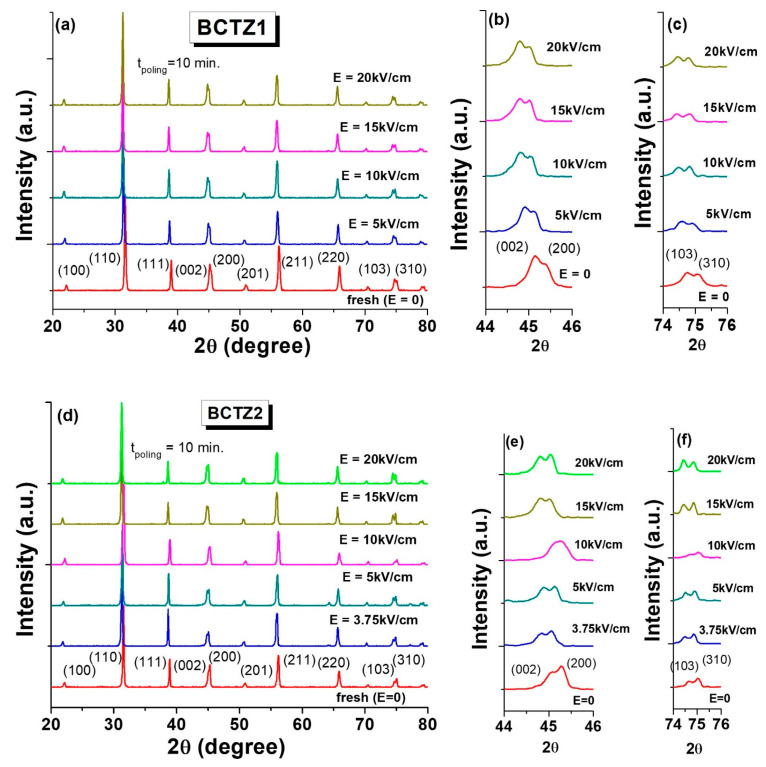
Experimental field-induced remanent structural modifications in BCTZ ceramics: (**a**–**c**) BCTZ1; (**d**–**f**) BCTZ2.

**Figure 8 materials-15-08809-f008:**
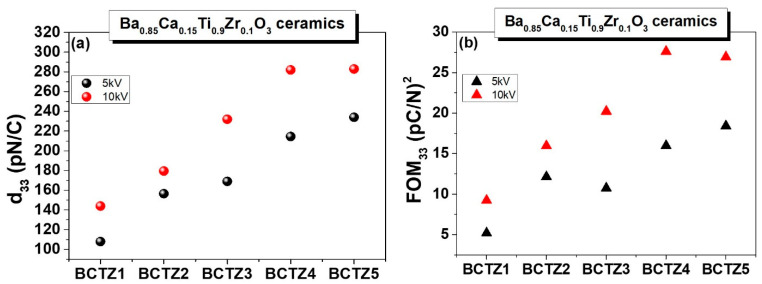
(**a**) Piezoelectric coefficient *d*_33_ and (**b**) *FOM*_33_ of BCZT ceramics determined for poling fields of 5 and 10 kV/cm.

**Table 1 materials-15-08809-t001:** Empirical parameters derived from the dielectric and ferroelectric data, the result of fitting with Curie–Weiss and Curie–Weiss-modified laws and relative density of BCTZ ceramics.

	Samples	*ε_m_*	*T_m_* (°C)	*T*_0_ (°C)	*C* × 10^5^ (°C)	*η*	*P_rem_* (µC/cm^2^) for *E_appl_* = 15 kV/cm	*E_c_* (kV/cm) for *E_appl_* = 15 kV/cm	Relative Density (r.d.)
1.	BCTZ1	6590	97.5	83	1.32	1.61	3.6	3	75%
2.	BCTZ2	10,620	92.5	92	1.10	1.60	4.35	2.4	76%
3.	BCTZ3	10,564	92.5	87	1.60	1.55	9	2.4	96%
4.	BCTZ4	11,400	87.5	88	1.46	1.66	10.1	2.3	95%
5.	BCTZ5	12,120	102.5	97	1.64	1.59	7.5	2.1	97%

**Table 2 materials-15-08809-t002:** Comparison of the electrocaloric properties of BCTZ with those of other Ba_0.85_Ca_0.15_ Zr_0.10_Ti_0.90_ (BCTZ) ceramics reported in the literature.

Ceramic	ΔT (K)	T(K)	ζ (Kmm/kV)	Ref.
BCTZ3	0.42	353	0.34	This study
BCTZ4	0.56	353	0.45	This study
BCTZ5	0.29	348	0.24	This study
Rod-like BCTZ	0.492	360	0.289	[48]
BCTZ	0.152	373	0.19	[49]
BCTZ	0.118	363	0.164	[52]
BCTZ	0.4	370	0.186	[53]

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
