# Peer review of "Optimization of Processing Steps for Superior Functional Properties of (Ba, Ca)(Zr, Ti)O_3_ Ceramics"

_materials, 2022, doi:10.3390/ma15248809_

Round 1

Reviewer 1 Report

The manuscript is based on the comprehensive study of BCZT prepared via different processing conditions and methods. The manuscript may be of interest to readers of this field. Therefore, I would like to recommend it for publication in materials after making the following changes/corrections/modifications.

1.       Overall, the language needs to be polished as there are many grammatical mistakes and typos.

2.       The experimental section should be revised as some information should be corrected such as time for milling etc.

3.       Although a detailed discussion on the XRD patterns has been provided but the identified and unidentified peaks must be marked in figure 1

4.       Such a considerable variation in the relative density is very unusual. From the SEM micrographs, it apparently looks like there is no huge difference in the porosity between 2d and 2e.

5.       Figure 3a shows a variation of relative permittivity with respect to a temperature which is almost consistent with the relative density. But Fig 3a also shows very strange behavior related to dielectric loss. In principle, with an increase in relative density, the dielectric loss should decrease.  Therefore, the authors should provide the reason for the anomalous behavior.

6.       What is the applied voltage for remanent polarization listed in table 1? It should be added.

7.       Why is such a low field used for PE measurements as these ceramics are expected to have a very high breakdown strength? I suggest performing measurements at a high electric field.

8.       The electrocaloric and piezoelectric results should be compared with the literature in a table form.

Author Response

Dear Reviewer,

Thank you for the overall observations. We have considered all the observations and we performed detailed corrections. All the modifications are indicated in the text in red color and the answers are shown below in blue.

  1. Overall, the language needs to be polished as there are many grammatical mistakes and typos.

Answer: We performed a detailed correction, and all the modifications are indicated in the text.

  1. The experimental section should be revised as some information should be corrected such as time for milling etc.

Answer: We added this information in the Experimental section part.

  1. Although a detailed discussion on the XRD patterns has been provided but the identified and unidentified peaks must be marked in figure 1

Answer: We marked in Fig. 1, a the small peaks which do not belong to the perovskite phase.

  1. Such a considerable variation in the relative density is very unusual. From the SEM micrographs, it apparently looks like there is no huge difference in the porosity between 2d and 2e.

Answer: The variation in density is caused by the differences in the processing steps (synthesis, calcination, milling steps and then, the sintering parameters). In fact, only BCTZ1 and BCTZ2 have a low relative density of 75% r.d. and 76% r.d., respectively and all the other ceramics have a high density. The Figures 2d and 2e show the micrographs in fracture of the ceramics BCTZ4 and BCTZ5 which indeed, have similar level of porosity: BCTZ4 (Fig. 2,d): 95% r.d. and BCTZ5 (Fig. 2,e): 97% r.d. The relative densities are mentioned in the text and have been introduced in the Figure 2 caption, as well.

  1. Figure 3a shows a variation of relative permittivity with respect to a temperature which is almost consistent with the relative density. But Fig 3a also shows very strange behavior related to dielectric loss. In principle, with an increase in relative density, the dielectric loss should decrease.  Therefore, the authors should provide the reason for the anomalous behavior.

Answer: When using fast sintering processes (as SPS), some ionic species maybe not be fully equilibrated in the system and the sintered ceramics may have oxygen vacancies locally compensated or not in the overall volume. All of these charges usually contribute to increase the dielectric losses by hoping mechanisms. We did not consider it necessary to provide a complete study of the dielectric relaxations in this paper, which might be studied in detail in a further study, by broadband dielectric spectroscopy in a large temperature range. It was added in the text: In the case of ultrafast sintering processes as SPS, some ionic species maybe not be fully equilibrated in the system and the presence of oxygen vacancies is likely, and such charges may contribute to the increase of the dielectric losses by hoping mechanisms [18]. This can explain the higher losses characteristic of the BCTZ5 ceramics with respect to the others observed in Fig. 2b.   

  1. What is the applied voltage for remanent polarization listed in table 1? It should be added.

Answer: The applied field for determining the remanent polarization is Eappl=15kV/cm, which was introduced in Table 1.

  1. Why is such a low field used for PE measurements as these ceramics are expected to have a very high breakdown strength? I suggest performing measurements at a high electric field.

Answer: The applied field of Eappl=15kV/cm was enough to obtain the saturation of the P(E) loops for all the BCTZ samples and to provide a relevant comparison. We did not propose to investigate the breakdown field in these compositions. This can be performed by high-field measurements with Weibull statistical analysis, but such a study is far from the topic of the present paper.

  1. The electrocaloric and piezoelectric results should be compared with the literature in a table form.

Answer: We added Table 2 in the manuscript with a comparison between the electrocaloric properties of our ceramics with others reported in the literature.

Reviewer 2 Report

The manuscript by Ciomaga et al. presents an interesting work on optimization of processing and electrical properties of BCZT ceramics with a MPB composition. The authors prepared five different samples, and investigated the dielectric, piezoelectric and electrocaloric properties. However, the manuscript has several serious flaws that will require much more work to make it publishable.

(1) In Figure 3a, the shift of Tm for different samples is more than 10 degree C. It means that the MPB composition has been changed during the sintering processes. Therefore, it does not make sense to compare the electrical properties of samples prepared with different composition and processing.      

(2) In Figure 1e, please check the labels of the peaks for BCTZ2. 

(3) The calcination temperature and time for BCTZ2, BCTZ3 and BCTZ4 are the same. However, the density of BCTZ2 is much lower than other samples. What happened during the densification?

(4) The ECE were calculated by using Maxwell relations. The P value was usually fitted by polynomial, so what's the polynomial order the authors used? 

(5) Concerning the electric field dependence of XRD patterns, peaks at (002)/(200) and (103)/(310) were slightly shifted. What might be the reason for this? 

(6) In this work, the reported d33 values are much smaller than that of MPB BCTZ ceramics in literature. It is clear to me that different processing steps do not help to optimize piezoelectric performance.

Author Response

Dear Reviewer,

Thank you for the overall observations. We have considered all the observations and we performed detailed corrections. All the modifications are indicated in the text in red color and the answers are shown below in blue.

(1) In Figure 3a, the shift of Tm for different samples is more than 10°C. It means that the MPB composition has been changed during the sintering processes. Therefore, it does not make sense to compare the electrical properties of samples prepared with different compositions and processing.     

 Answer: The starting nominal composition of the present BCTZ ceramics was the same, the full incorporation of all the precursors by XRD confirmed the chemical reaction, and the formation of other secondary phases is below the XRD detection limit and therefore, there are no reasons to consider deviations from stoichiometry or from the overall nominal compositions in the present ceramics. We wanted to show in this paper that even the composition was the same, the different processing routes may provide local structural and maybe compositional heterogeneity which resulted in differences in the microstructural parameters (density, grain size) and phase composition which gives variations of the properties. Therefore, we think it makes sense to compare the resulting properties for a given specific composition. Variations in the permittivity maximum or in the permittivity vs. temperature dependences in ferroelectric ceramics with the same composition around MPB have been also found in the literature [Su, Y.; Weng, G. The shift of Curie temperature and evolution of ferroelectric domain in ferroelectric crystals. J. Mech. Phys. Solids 2005, 53, 2071–2099; Bao, Y.; Huang, B.; Zhou, K.; Roscow, J.; Bowen, C. Hierarchically structured lead-free barium strontium titanate for low-grade thermal energy harvesting. Ceram. Int. 2021, 47, 18761–18772; Pu, Y.; Zhu, J.; Zhu, X.; Luo, Y.; Li, X.; Wang, M.; Jing, L.; Li, X.; Zhu, J.; Xiao, D. Enhanced Ferroelectric Properties of Intragranular-Porous Pb(Zr0.95Ti0.05)O3Ceramic Fabricated with Carbon Nanotubes. J. Am. Ceram. Soc. 2010, 93, 642–645; X. Zhou, Zhou, K.; Zhang, D.; Bowen, C.; Wang, Q.; Zhong, J.; Zhang, Y. Perspective on Porous Piezoelectric Ceramics to Control Internal Stress, Nanoenergy Adv. 2022, 2, x. https://doi.org/10.3390/nanoenergyadv2040014].

(2) In Figure 1e, please check the labels of the peaks for BCTZ2. 

Answer: We have checked and corrected the label.

(3) The calcination temperature and time for BCTZ2, BCTZ3 and BCTZ4 are the same. However, the density of BCTZ2 is much lower than other samples. What happened during the densification?

Answer: We wanted to show in this comparative study that also the milling type and parameters play a role in the phase formation and properties and not only the calcination and sintering temperature and time, which were presented by other authors in literature. BCTZ2 was produced in a different way with respect to the other ceramics, as clearly described in the text: wet ball milling of reagents for 30 min. at 350 rpm in a planetary mill (RETSCH S100), followed by a calcination step at 1000°C for 4 h…  It seems that this type of milling provides a less effective densification than the other methods and if one wants to improve the densification by using this specific method, it should perform an extensive study dedicated to this milling method and optimize the available parameters: time, rpm, intermediate calcination-milling steps, etc.  

(4) The ECE were calculated by using Maxwell relations. The P value was usually fitted by polynomial, so what's the polynomial order the authors used? 

Answer: We used a fourth-order polynomial to fit the experimental data.

(5) Concerning the electric field dependence of XRD patterns, peaks at (002)/(200) and (103)/(310) were slightly shifted. What might be the reason for this? 

Answer: These shifts may be assigned to field-induced stress modifications at different applied fields, which are related again with the initial polymorph composition. Detailed structural calculations may provide data concerning the internal stresses as a function of the applied field, but high-energy XRD is necessary for this.

(6) In this work, the reported d33 values are much smaller than that of MPB BCTZ ceramics in literature. It is clear to me that different processing steps do not help to optimize piezoelectric performance.

Answer:  Indeed in the literature, there are presented high piezoelectric response for BCTZ materials, but concerning the synthesis methods of producing the dense and porous BCTZ ceramics, also there are reported data of piezoelectric responses for the same BCTZ composition in which the authors obtained similar or even low values for BCTZ ceramics. We have added and completed in the text references which present the same values as d33.

Round 2

Reviewer 1 Report

The manuscript may now be accepted. 

Reviewer 2 Report

The authors have properly answered the questions from the reviewers, and the manuscript is suitable for publication.